# Exploring Optimal Transport for Event-Level Anomaly Detection at the Large Hadron Collider

Nathaniel Craig[1,2], Jessica N. Howard[2*] and Hancheng Li[1]

**1** Department of Physics, University of California, Santa Barbara, CA USA
**2** Kavli Institute for Theoretical Physics, Santa Barbara, CA USA
* jnhoward@kitp.ucsb.edu

July 12, 2024

## Abstract

**Anomaly detection is a promising, model-agnostic strategy to find physics beyond the Standard Model. State-of-the-art machine learning methods offer impressive performance on anomaly detection tasks, but interpretability, resource, and memory concerns motivate considering a wide range of alternatives. We explore using the 2-Wasserstein distance from optimal transport theory, both as an anomaly score and as the input to interpretable machine learning methods, for event-level anomaly detection at the Large Hadron Collider. The choice of ground space plays a key role in optimizing performance. We comment on the feasibility of implementing these methods in the L1 trigger system.**

# 1 Introduction

In the search for new physics Beyond the Standard Model (BSM), it is becoming increasingly important to consider model-agnostic search strategies in addition to standard, dedicated searches for a particular model of new physics. The goal of such broad strategies is to reliably recognize any anomalous data signature which deviates from the Standard Model (SM) baseline. These anomaly detection strategies could then be applied to real data, either at the trigger level or in offline analyses, to look for many BSM models at once. This is appealing given the plethora of possible BSM signatures and the immense resource requirements of dedicated searches.

Anomaly detection, fueled by state-of-the-art Machine Learning (ML) methods and several Large Hadron Collider (LHC) data challenges [1–3], has increasingly grown in popularity since the discovery of the Higgs boson [4, 5] over a decade ago. These methods are even starting to be adopted in experimental searches and trigger algorithms [6–14]. For in-depth reviews see Refs. [15–17]. When designing anomaly detection algorithms, the types of LHC data considered can broadly be split into three categories. The first considers analyzing the 4-momenta of jet constituents individually or as calorimeter deposits, while the second considers high-level, reconstructed quantities such as invariant masses and jet substructure variables. The third operates at the event level considering the 4-momenta of reconstructed objects and may either correspond to offline or online, trigger-level analyses. Numerous ML methods have been specially developed, or adapted from classification tasks, for anomaly detection. Typically, the majority of anomaly detection methods are unsupervised, so that they may be trained on real, control data, thereby side-stepping biases from simulations. However, many methods instead fall under the umbrella of less-than-supervised [15] (i.e. semi-supervised or weakly supervised). Additionally, some works [18, 19] have employed supervised learning in an unsupervised setting. Instead of using simulated signal data, they instead transform background data (either real or simulated) so that it appears signal-like. This "anomaly augmented" background data is then used as the signal data for supervised training. In general, ML-based methods offer impressive performance on anomaly detection tasks, far surpassing standard anomaly detection methods, such as kinematic or particle multiplicity cuts [15, 17].

However, this improved performance comes with some drawbacks. For example, autoencoders, one of the most popular unsupervised anomaly detection methods first used in high-energy contexts in Refs. [20, 21], have been shown to have topological obstructions [22], are prone to high false-negative rates [23], and have difficulty identifying low-complexity anomalies [24, 25]. Much work has gone into addressing such concerns and developing unbiased, ML-based anomaly detection algorithms [6, 15, 17, 25–33]. However, given the opaque nature of ML methods, the risk of additional, unknown failure modes may still remain. Moreover, despite recent progress with autoencoder methods [8, 9], the ease of deploying state-of-the art ML methods, particularly in an online setting where storage space and computing resources are limited, remains a concern [1]. Furthermore, for many ML methods, it can be difficult to interpret learned relationships and quantify uncertainties [17].[1] It is therefore useful to explore alternative strategies such as new representations of data that facilitate anomaly detection, either in conjunction with interpretable ML algorithms or without ML altogether.

The theory of Optimal Transport (OT) offers a variety of such promising representations. OT theory is a branch of mathematics which defines a distance metric between probability distributions. Given that the core idea of anomaly detection is to differentiate between SM and BSM distributions over detector signatures, using OT distances as an anomaly score (or as

---

[1]We note that recent work [28] has made great strides towards addressing uncertainty quantification for a particular ML model. Their model ELUQuant uses normalizing flows to estimate the posteriors of a physics-informed Bayesian Neural Network and is capable of capturing both aleatoric and epistemic uncertainties.

input to distance-based ML algorithms) is sensible. Unlike density estimation strategies [15], OT does not need to learn a probability model of the data since it is predisposed to detect distributional differences. Therefore, using OT distances as an anomaly score does not require training, is directly interpretable, and is amenable to error analysis. To-date, various OT distances have been widely used in LHC classification tasks [34–43]. Some previous works have also explored using OT for anomaly detection on jet constituent [44, 45] and high-level variable [40] data. Ref. [44] investigated using p-Wasserstein distances to augment the latent space of autoencoders, directly as an anomaly score, and in combination with interpretable ML methods. Ref. [45] created a neural embedding of the Energy Mover's Distance (a modified version of the 1-Wasserstein distance) between jets. Ref. [40] modified the ground space of the Energy Mover's Distance and used it as an anomaly score on high-level variables.

In this work, we investigate using balanced OT for anomaly detection on L1 trigger-level data from the CMS anomaly detection challenge dataset (CMS ADC 2021) [1]. Because of its advantageous mathematical properties [35], we focus on the 2-Wasserstein distance. We use it both directly as an anomaly score as well as in tandem with two interpretable ML techniques: One-class Support Vector Machines (SVMs) [46] and k-Nearest Neighbors (kNNs) [47] trained using anomaly augmented background as signal. Thus, the methods we consider range from unsupervised to weakly supervised. We find that, unsurprisingly [48], the choice of ground space plays a large role in performance. By considering a different ground space from previous works [35–38, 41–43], we are able to significantly improve performance. We compare our results to baseline methods and a recent state-of-the-art ML method [18] benchmarked on the same dataset. We conclude with comments about the feasibility of incorporating these methods into the Level-1 (L1) trigger system and directions for future work to improve performance.

## 2 Dataset

In this work we use simulated event-level anomaly data from the CMS anomaly detection challenge dataset (CMS ADC 2021) [1], which considers proton-proton collisions at a center-of-mass-energy of 13 TeV simulated using PYTHIA 8.240 [49], and DELPHES 3.3.2 [50] with the Phase-II CMS detector card. There are six sub-datasets: a background dataset comprising only SM processes, four signal datasets with potential BSM processes for new particles, and a "mystery" dataset containing both background and signal events for an unknown signal process. In this work, we will not consider the latter.

This dataset is restricted so that any event can have at most 4 electrons/photons,[2] 4 muons, and 10 jets. There is also one aggregate missing transverse energy (MET) object per event. For each event, the $p_\mathrm{T}/\mathrm{GeV}$, $\eta$, $\phi$, and identity of each object is stored in a [19,4], zero-padded array. Events are also required to have at least one electron/photon or muon with $p_\mathrm{T} > 23$ GeV. Additionally, all electrons/photons must have $p_\mathrm{T} > 3$ GeV and $|\eta| < 3$; all muons must have $p_\mathrm{T} > 3$ GeV and $|\eta| < 2.1$; all jets must have $p_\mathrm{T} > 15$ GeV and $|\eta| < 4$ and are reconstructed using FASTJET [51] with the anti-$k_t$ algorithm [52]. This data format is meant to mimic the kind of information available at the LHC's L1 trigger level, where bandwidth, latency, and resources are limited.

The SM background data contains four leading processes with representative fractions

---

[2]The dataset does not distinguish between electrons and photons.

determined by their leading-order cross section and trigger efficiency:

$$pp \rightarrow W^{\pm} + \text{jets} \rightarrow l^{\pm} \nu_l + \text{jets} \qquad (59.2\%)$$
$$pp \rightarrow \text{jets} \qquad (33.8\%)$$
$$pp \rightarrow Z + \text{jets} \rightarrow l^+ l^- + \text{jets} \qquad (6.7\%)$$
$$pp \rightarrow t\bar{t} + \text{jets} \qquad (0.3\%), \qquad (1)$$

where $l = e, \mu, \tau$. Four BSM signal models are also considered:

- **Neutral scalar boson** $A$: The mass of this BSM particle is $m_A = 50$ GeV. The production mechanism is $pp \rightarrow A + X \rightarrow Z^* Z^* + X$, $Z^* \rightarrow l^+ l^-$, where $X$ indicates inclusive activity.

- **Scalar boson** $h^0$: The mass of this BSM particle is $m_{h^0} = 60$ GeV. The production mechanism is $pp \rightarrow h^0 + X \rightarrow \tau^+ \tau^- + X$, where $X$ indicates inclusive activity.

- **Charged scalar** $h^{\pm}$: The mass of this BSM particle is $m_{h^{\pm}} = 60$ GeV. The production mechanism is $pp \rightarrow h^{\pm} + X \rightarrow \tau^{\pm} \nu + X$, where $X$ indicates inclusive activity.

- **Leptoquark ($LQ$)**: The mass of this BSM particle is $m_{LQ} = 80$ GeV. The production mechanism is $pp \rightarrow LQ \rightarrow \tau b$.

As was pointed out in Ref. [18, 53], many standard methods are sensitive to these BSM signals. Therefore, in order to establish baseline performance, we consider standard cuts on kinematic quantities: $p_{\mathrm{T}}$, MET, and multiplicities of $e/\gamma$, $\mu$, jet, respectively. The performance of baseline quantities are reported in Table 2 of Appendix A.

## 2.1 Anomaly augmentations

In addition to fully unsupervised methods, we also consider using supervised learning in an unsupervised setting. To accomplish this, we apply the anomaly augmentations described in Ref. [18] to the SM background data described in the previous section. This creates a "fake" signal dataset which we can subsequently use in supervised training.

We consider three types of anomaly augmentations [18]:

1. **Multiplicity increase:** This transformation randomly adds some number of $e/\gamma$, $\mu$, and jets to each event in the range $(n_{e/\gamma}, 4 - n_{e/\gamma})$, $(n_\mu, 4 - n_\mu)$, $(n_{\mathrm{jets}}, 10 - n_{\mathrm{jets}})$, respectively. The $p_T$ of each new particle is a randomly chosen fraction of the highest $p_T$ of any object in that event added to the base required by that object's selection criteria. The $\eta, \phi$ of each object are uniformly chosen within the allowed limits corresponding to that object's type. After all objects are added, the MET of the event is recalculated.

2. **Multiplicity increase, with constant** MET **and** $p_{\mathrm{T}}$**:** This transformation keeps the total $p_{\mathrm{T}}$ and MET constant while still increasing the overall object multiplicity. This is achieved by splitting an existing object to create two new objects. The combined $p_T$ of the two new objects is equal to the original. The $\eta, \phi$ of each new object are then randomly smeared with Gaussian noise.

3. MET **and** $p_{\mathrm{T}}$ **shift:** This transformation randomly shifts the a) MET, b) reconstructed object $p_T$, or c) both by a constant multiplicative factor. Transformations (a),(b),(c) are chosen with equal probability. To satisfy selection criteria, the multiplicative factor for shifting $p_T$ is chosen uniformly in the range $[1, 5]$ (i.e. the $p_T$ of objects will never be down-shifted such that it might violate an object's minimum $p_T$ criteria). No such restriction exists for MET, so its multiplicative factor is chosen uniformly in the range $[0.5, 5]$.

In general, we apply augmentations (1), (2), and (3) with equal probability. However, there are certain events for which augmentation (2) could not transform the event without causing the event selection criteria to be violated (approximately 8%). For these events, we instead apply augmentations (1) and (3) with equal probability. Taking this into account, approximately 37.3%, 29.3%, 37.3% of events are transformed with augmentations (1), (2), and (3), respectively.

## 3 Methods

In this section, we describe the details of the methods used in this work. We utilize the Python Optimal Transport (POT) library [54] to calculate 2-Wasserstein distances and the SCIKIT-LEARN library [55] to implement the k-Nearest Neighbor and one-class Support Vector Machine algorithms.[3]

### 3.1 Optimal Transport (OT) distance

OT theory provides a framework for comparing probability distributions by assigning a cost to transform one probability distribution into another. The cost of moving a differential piece of the source distribution to construct the target distribution is the amount of probability mass weighted by (an integer power of) how far that differential piece has to move in the ground space. The collection of these differential movements is called a transport plan, and the net cost of executing the *optimal* (lowest-cost) transport plan induces a distance between the source and target distributions. In this way, OT theory gives a way to elevate cost metrics defined on the ground space to a metric between probability distributions defined over that ground space.

A mathematically robust choice of ground space cost metric is the squared Euclidean distance, $c(x, x') = |x - x'|^2$. The distance associated with the solution of the OT problem with this cost metric is the 2-Wasserstein distance. Let $\mathcal{E}$ ($\mathcal{E}'$) be a discrete source (target) distribution over a $d$-dimensional ground space, $X$. Therefore, $\mathcal{E}$ ($\mathcal{E}'$) is a collection of probability point masses, $p_i$ ($p'_j$), at locations $x_i \in X$ ($x'_j \in X$). The 2-Wasserstein distance is then given by

$$W_2(\mathcal{E}, \mathcal{E}') = \left( \min_{g \in \Gamma(\mathcal{E}, \mathcal{E}')} \sum_{i,j} g_{ij} |x_i - x'_j|^2 \right)^{1/2} \qquad (2)$$

where $\Gamma(\mathcal{E}, \mathcal{E}')$ is the set of possible joint distributions with marginals $\mathcal{E}, \mathcal{E}'$, respectively. The $g^* \in \Gamma(\mu, \nu)$ which minimizes the above is the optimal transport plan.

While many works have employed variations on the 1-Wasserstein distance (or *Earth Mover's Distance*) due to its conceptual simplicity [34, 37, 38, 40–43], using the alternative 2-Wasserstein structure is mathematically beneficial for several reasons. The strictly convex cost function guarantees that $g^*$ is unique, admits unique geodesics between distributions, and lends itself to efficient linearization algorithms leveraging the resulting pseudo-Riemannian structure [36]. Therefore, in this work, we will only consider the 2-Wasserstein distance. Having fixed this choice, we are left to define the underlying ground space, $X$, and the distributions over it.

The key expectation is that the 2-Wasserstein distance between two SM events should generally be less than the distance between a SM event and a BSM event. Thus, we can attempt to use the 2-Wasserstein distance between a test event and a representative ensemble of SM events to derive an anomaly score. More precisely, let $\mathcal{T}$ be a test event of unknown type and

---

[3]The code to reproduce all results and figures in this work is publicly available at https://github.com/hancheng-li/anomaly_detection_code.

$\{\mathcal{S}_1, ..., \mathcal{S}_N\}$ be a collection of $N$ SM reference events. This will, in turn, yield $N$ 2-Wasserstein distance values, $\{W_2(\mathcal{T}, \mathcal{S}_1), ..., W_2(\mathcal{T}, \mathcal{S}_N)\}$. We then want to condense these $N$ event-to-event distances into a single event-to-ensemble distance. There are many ways one could envision doing this. Ref. [44] explored taking the average value of $\{W_2(\mathcal{T}, \mathcal{S}_1), ..., W_2(\mathcal{T}, \mathcal{S}_N)\}$, as well as choosing the reference events $\{\mathcal{S}_1, ..., \mathcal{S}_N\}$ to be the $k$-medoids of the SM distribution (i.e. reducing $N \rightarrow k$) and taking the minimum of $\{W_2(\mathcal{T}, \mathcal{S}_1), ..., W_2(\mathcal{T}, \mathcal{S}_k)\}$. In this work, we choose the anomaly score for test event $\mathcal{T}$ to be the minimum of $\{W_2(\mathcal{T}, \mathcal{S}_1), ..., W_2(\mathcal{T}, \mathcal{S}_N)\}$. Using the minimum distance between a test event and an ensemble of SM events as an anomaly score is consistent with the expectation that a given SM event should be closer to at least a subset of the SM background events than a corresponding BSM event, even if the background sample is highly inhomogenous. We have verified that using the maximum distance as an anomaly score is ineffective precisely because of the inhomogeneity of the background sample. Similarly, we have verified that the average distance also suffers from the inhomogeneity of the background sample.

Given that probability distributions are sensitive to the choice of initial observables (ground space) and coordinate transformations [48], the 2-Wasserstein distance is sensitive as well. This means that selecting a ground space that maximizes the differences between SM and BSM events (and minimizes differences between SM events) is essential. Past works [35–38, 41–44] have opted for a physically-motivated ground space where the ground space is the relative $\eta, \phi$ plane centered on a jet's 4-vector location. The (discrete) probability distribution is then the $p_T$ associated to each jet constituent in this plane. We call this choice of the $\eta, \phi$ plane the "2D" ground space.

However, this intuitive picture has drawbacks when considering scaling OT methods from a jet-level to an event-level picture. For balanced OT, distributions must be identically normalized and thus information about the total $p_T$ associated with an event is lost. While this information may be regained by moving to an unbalanced OT framework [35, 40], the many possible unbalanced OT prescriptions come with varying degrees of mathematical well-posedness and typically entail a significant increase in computational cost [35]. Additionally, the underlying physical symmetries of collider events motivate aligning distributions in the $\eta, \phi$ plane of the detector before computing OT distances. Unlike with jets, accounting for such symmetries by pre-processing the data is typically infeasible for full events, causing physically identical distributions to appear different. Considering alternative ground spaces may offer a way to circumvent or minimize the effect of these difficulties. For example, Ref. [34] sought to address many of these shortcomings by considering a spectral ground space that encodes pairwise particle angles and products of particle energies.

In this work, we consider an alternate "3D" ground space which adds an additional $p_T/\text{GeV}$ dimension to the standard $\eta, \phi$ plane. To form a distribution over this space, each of the 19 objects is given an equal probability weighting. Non-existent objects (i.e. zero padded rows in an event), therefore, sit at the origin. This choice of a less-intuitive ground space indirectly restores information about the total $p_T$ in an event, which is crucial for detecting anomalies.[4] It also allows us to use the 2-Wasserstein distance in a balanced OT setting, which maintains its mathematical advantages and admits computationally efficient algorithms. When evaluating the 2D and 3D OT distances as anomaly scores we construct a test set from $1,000$ SM background events and $1,000$ BSM signal events (for each signal model).

---

[4]We note that this choice of presentation is analogous to that used by many state-of-the-art machine learning methods.

## 3.2   k-Nearest Neighbors (kNNs) using OT distances

The kNN algorithm [47] provides a simple and interpretable supervised machine learning strategy for binary classification. Given a training set of events from both classes (SM background and BSM signal), a test event's class is determined by a majority vote from its $k$ nearest neighbors, where $k$ is a hyperparameter. The 2-Wasserstein distance is used to calculate the distance between events to determine which neighbors are closest.

Using the 2-Wasserstein distance in this way helps take advantage of the tendency for events to cluster instead of fully separate [35, 36, 38, 44]. The SM background events in our study are highly inhomogeneous, in that they include many different physics processes under the same SM background label. Therefore, it is possible that the differences between two SM subprocesses are more extreme than the differences between a SM subprocess and a BSM process. In this scenario, a clean separation would be unlikely. However, we would still expect events from the same process to be more similar to each other than to events from a different process. This would lead to clusters of events forming in this abstract 2-Wasserstein space. This has been empirically observed in prior works operating on the 2D ground space [35,36,38,44].

Even though kNNs are a supervised classification algorithm, we can promote them to a weakly supervised anomaly detection method by using anomaly augmented SM events (instead of BSM events) as "signal" during training. We employ the anomaly augmentations proposed in Ref. [18], see Section 2.1 for details. We train on a random mixture of $1,000$ SM background events and $1,000$ (statistically independent) anomaly augmented background events. We reserve 25% of this set for validation; we scan over the hyperparameter $k$ in the range $[5, 495]$ in increments of 10 and select the value which yields the best performance. We then construct a test set with a random mixture of $1,000$ SM background events and $1,000$ BSM signal events (for each signal model).

Additionally, to gauge how much information is contained in the representations of the data that we consider (i.e. 2D and 3D ground space) we also perform fully supervised classification using kNNs. In some sense, this is a best-case performance limit for anomaly detection, and serves to investigate how the choice of ground space affects performance. For each BSM signal model, we train on a random mixture of $1,000$ SM background events and $1,000$ BSM signal events, with 20% reserved for validation and 20% reserved for testing. We scan over the hyperparameter $k$ in the range $[5, 495]$ in increments of 10 and select the value which yields the best performance. For each classification task, the mean and standard deviation of the results are estimated via 5-fold cross-validation.

## 3.3   One-class Support Vector Machines (SVMs) using OT distances

One-class SVMs [46] are another simple and interpretable machine learning strategy. Unlike their vanilla counterpart (SVMs), one-class SVMs are fully unsupervised and operate by drawing a decision boundary around the training SM data. It therefore implicitly learns to estimate the density of SM events and assign a BSM label to test events falling in low density regions. Therefore, one-class SVMs are commonly used for "outlier" detection tasks.[5] Similar to the kNN case, we distribute events in a high-dimensional, abstract 2-Wasserstein space where we expect like-events to cluster. There are two hyperparameters: $\nu$, which controls the margin of the decision boundary, and $\gamma$ which is (inversely) related to the variance of the Gaussian kernel, $K(\mathcal{E}, \mathcal{E}') = \exp\left(-\gamma W_2(\mathcal{E}, \mathcal{E}')^2\right)$.

We train on a set of $10,000$ SM background events. The increase in training statistics is because we found $1,000$ to be slightly too small of a sample to draw an accurate descision

---

[5]While outliers are a certain type of anomaly it has been previously noted that, in high-energy physics contexts, it is more likely for anomalies to be "over-densities" [3]. This observation generally makes outlier detection methods less favorable for anomaly detection at the LHC.

| | | 2D OT | 3D OT | 3D OT+anomaly kNN $k = 15$ | 3D OT+one-class SVM $\nu = 0.2, \gamma = 0.35$ |
|---|---|---|---|---|---|
| AUC | $A$ | $0.6544 \pm 0.01119$ | $\mathbf{0.8370 \pm 0.008752}$ | $0.8324 \pm 0.005852$ | $0.7682 \pm 0.00621$ |
| | $h^0$ | $0.6151 \pm 0.007914$ | $\mathbf{0.7418 \pm 0.008213}$ | $0.6864 \pm 0.008349$ | $0.6622 \pm 0.007407$ |
| | $h^\pm$ | $0.6392 \pm 0.002815$ | $\mathbf{0.9129 \pm 0.005798}$ | $0.8134 \pm 0.006918$ | $0.8096 \pm 0.008206$ |
| | $LQ$ | $0.6577 \pm 0.008992$ | $\mathbf{0.8410 \pm 0.007098}$ | $0.7570 \pm 0.007815$ | $0.7386 \pm 0.01378$ |
| $\epsilon_b^{-1}$ ($\epsilon_s = 0.3$) | $A$ | $5.180 \pm 0.6198$ | $17.68 \pm 3.140$ | $\mathbf{35.44 \pm 5.899}$ | - |
| | $h^0$ | $4.573 \pm 0.3412$ | $\mathbf{17.81 \pm 3.977}$ | $10.62 \pm 1.754$ | - |
| | $h^\pm$ | $4.506 \pm 0.3166$ | $\mathbf{63.40 \pm 11.01}$ | $22.12 \pm 3.928$ | - |
| | $LQ$ | $4.823 \pm 0.3244$ | $\mathbf{18.59 \pm 3.128}$ | $13.38 \pm 0.9646$ | - |
| SI ($\epsilon_s = 0.3$) | $A$ | $0.6814 \pm 0.03989$ | $1.255 \pm 0.1121$ | $\mathbf{1.776 \pm 0.1497}$ | - |
| | $h^0$ | $0.6410 \pm 0.02347$ | $\mathbf{1.258 \pm 0.1353}$ | $0.9741 \pm 0.07812$ | - |
| | $h^\pm$ | $0.6363 \pm 0.02211$ | $\mathbf{2.373 \pm 0.1989}$ | $1.404 \pm 0.1262$ | - |
| | $LQ$ | $0.6583 \pm 0.02165$ | $\mathbf{1.288 \pm 0.1093}$ | $1.096 \pm 0.03933$ | - |
| Max F1 ($\epsilon_s$) | $A$ | $0.7069 \pm 0.003794 \ (0.90)$ | $\mathbf{0.7947 \pm 0.005469 \ (0.86)}$ | $0.7608 \pm 0.007907 \ (0.80)$ | $0.7453 \pm 0.006904 \ (0.70)$ |
| | $h^0$ | $0.6830 \pm 0.005612 \ (0.94)$ | $\mathbf{0.7096 \pm 0.007947 \ (0.91)}$ | $0.6806 \pm 0.003511 \ (0.94)$ | $0.6742 \pm 0.008316 \ (0.72)$ |
| | $h^\pm$ | $0.7056 \pm 0.003958 \ (0.94)$ | $\mathbf{0.8459 \pm 0.01099 \ (0.87)}$ | $0.7626 \pm 0.005405 \ (0.89)$ | $0.7787 \pm 0.009127 \ (0.70)$ |
| | $LQ$ | $0.7117 \pm 0.005028 \ (0.92)$ | $\mathbf{0.7908 \pm 0.003962 \ (0.90)}$ | $0.7224 \pm 0.005807 \ (0.88)$ | $0.7240 \pm 0.01442 \ (0.71)$ |
| Max SI ($\epsilon_s$) | $A$ | $1.120 \pm 0.01180 \ (0.90)$ | $1.637 \pm 0.07674 \ (0.73)$ | $\mathbf{1.926 \pm 0.07768 \ (0.51)}$ | $1.703 \pm 0.0216 \ (0.68)$ |
| | $h^0$ | $1.043 \pm 0.01283 \ (0.90)$ | $\mathbf{1.267 \pm 0.1241 \ (0.22)}$ | $1.054 \pm 0.02054 \ (0.71)$ | $1.130 \pm 0.02716 \ (0.70)$ |
| | $h^\pm$ | $1.112 \pm 0.01125 \ (0.90)$ | $\mathbf{2.619 \pm 0.2490 \ (0.44)}$ | $1.492 \pm 0.09023 \ (0.45)$ | $2.425 \pm 0.1141 \ (0.68)$ |
| | $LQ$ | $1.132 \pm 0.01506 \ (0.89)$ | $\mathbf{1.593 \pm 0.09023 \ (0.61)}$ | $1.253 \pm 0.04667 \ (0.56)$ | $1.463 \pm 0.06187 \ (0.68)$ |

Table 1: Performance of the anomaly detection methods considered in this work for each signal dataset. The performance metrics considered are the area under the ROC curve (AUC), inverse background efficiency rate ($\epsilon_b^{-1}$), and Significance Improvement (SI). Where the latter two are reported at a signal efficiency rate of $\epsilon_s = 0.3$. Note that the 3D OT+one-class SVM case only gives binary label predictions which means its efficiency is only defined at one $\epsilon_s$. We also report the maximum SI and F1 values and their corresponding $\epsilon_s$ in the range $\epsilon_s \in [0.2, 1]$. A lower signal efficiency threshold ($\epsilon_s = 0.2$) is imposed when finding the maximum SI value, since the SI curve becomes unstable for low signal efficiencies.

boundary in this fully unsupervised setup. However, empirically for this dataset, we found that performance did not significantly change for $3,000$ training events or more. Our validation set is an equal, randomized mixture of SM background and all four BSM signals totaling $5,000$ events. We then construct a test set with a random mixture of $1,000$ SM background events and $1,000$ BSM signal events (for each signal model). We fix $\nu = 0.2$ and scan over $\gamma$ in the range $[0.05, 0.55]$ in increments of $0.1$. We then construct a test set with a random mixture of $1,000$ SM background events and $1,000$ BSM signal events (for each signal model).

## 4 Results

We compare performance across different methods with Receiver Operating Characteristic (ROC) curves which plot the the background rejection rate ($1 - \epsilon_b$) as a function of signal efficiency ($\epsilon_s$), as well as several standard metrics derived from ROC curves. Namely, we consider the area under the ROC curve (AUC), inverse background efficiency rate ($\epsilon_b^{-1}$), Significance Improvement (SI), and F1 score. All anomaly detection methods are evaluated on 5 separate test sets and the resulting ROC curves and metrics are averaged and standard deviation calculated. Figure 1 plots the mean and ±1-standard deviation error band of the ROC curves for all OT-based anomaly detection methods considered. Figure 2 plots the mean and ±1-standard deviation error band of the SI curves for all OT-based anomaly detection methods considered. Note that a lower signal efficiency threshold ($\epsilon_s = 0.20$) is imposed since the SI curve becomes unstable for low signal efficiencies. Table 1 reports the mean and standard deviation of all derived metrics.

A number of salient features are apparent. The first is the relatively poor performance

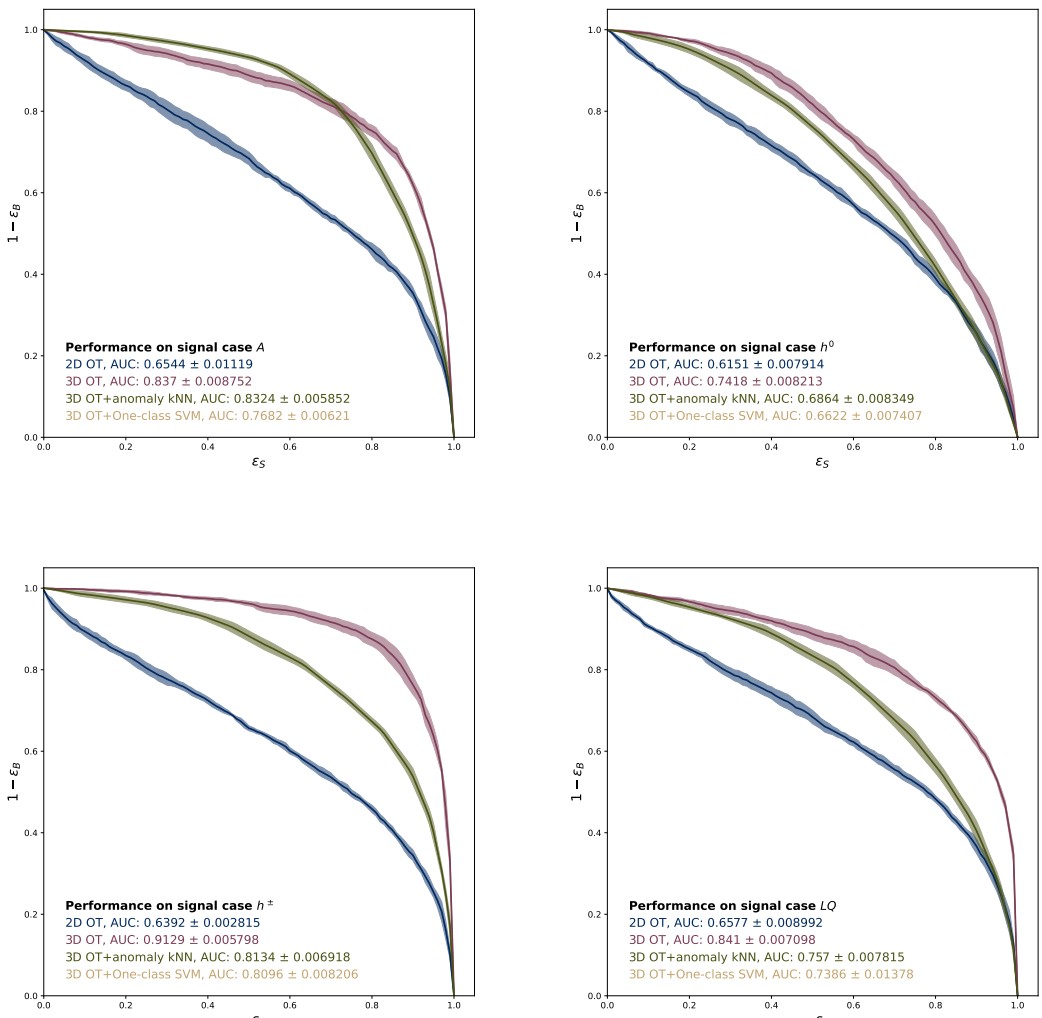

Figure 1: ROC curves with corresponding AUC values for each signal case are reported for the anomaly detection methods considered. We consider using the minimum OT distance as an anomaly score with two underlying ground spaces (2D OT, 3D OT) as well as using OT distances in tandem with interpretable machine learning methods (3D OT+anomaly kNN, 3D OT + one-class SVM). The 2D OT, 3D OT, and 3D OT + one-class SVM are all unsupervised methods whereas the 3D OT+anomaly kNN is weakly supervised. Note that the 3D OT+one-class SVM case only gives binary label predictions which means its efficiency is only defined at one $\epsilon_s$, thus its ROC curve is trivial and not shown.

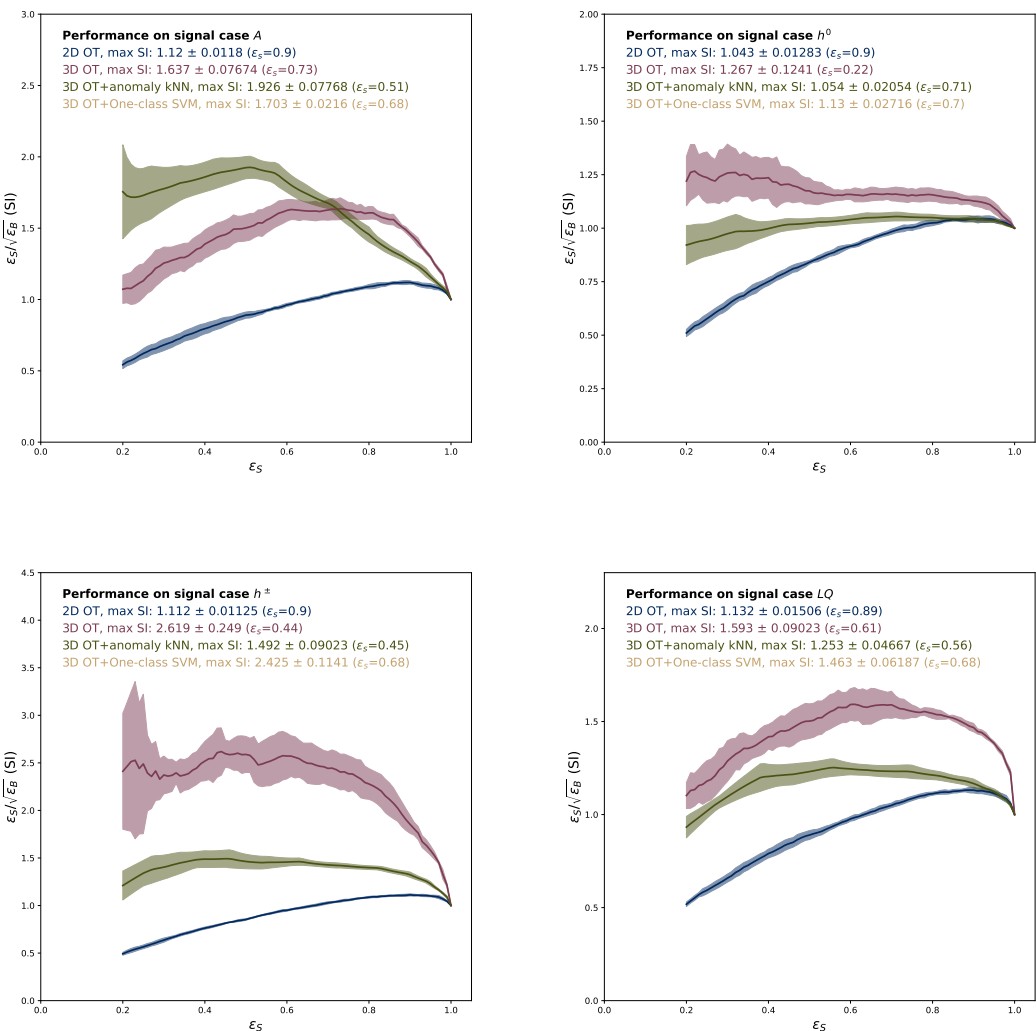

Figure 2: SI curves with corresponding max SI values for each signal case are reported for the anomaly detection methods considered. We consider using the minimum OT distance as an anomaly score with two underlying ground spaces (2D OT, 3D OT) as well as using OT distances in tandem with interpretable machine learning methods (3D OT+anomaly kNN, 3D OT + one-class SVM). The 2D OT, 3D OT, and 3D OT + one-class SVM are all unsupervised methods whereas the 3D OT+anomaly kNN is weakly supervised. A lower signal efficiency threshold ($\epsilon_s = 0.2$) is imposed since the SI curve becomes unstable for low signal efficiencies. The max SI value is found in this signal efficiency range (i.e. $\epsilon_s \in [0.2, 1]$). Note that the 3D OT+one-class SVM case only gives binary label predictions which means its efficiency is only defined at one $\epsilon_s$, thus its SI curve is trivial and not shown.

of anomaly detection using 2D OT distances alone, which is not competitive with either the total $p_T$ or total multiplicity baseline methods for the four signal models. This is not at all surprising, insofar as the total $p_T$ is a useful discriminant between SM and BSM events in this dataset, and is entirely absent from the balanced 2-Wasserstein distance between events using the 2D $\eta, \phi$ ground space.

The inclusion of constituent $p_T$ information in the 3D OT distance leads to dramatically improved performance in anomaly detection. This is particularly apparent for the $h^\pm$ signal, which sees the greatest increase in performance; the $h^\pm$ signal has a heavy total $p_T$ tail, reflected by the efficacy of the total $p_T$ baseline selection for this signal model. More broadly, the performance of the 3D OT distance is comparable to the best baseline method for each signal model, despite the fact that the best baseline method varies between total $p_T$ and total multiplicity from model to model (see Table 2). With the exception of the $A$ signal model (discussed further below), anomaly detection using the 3D OT distance begins to approach the performance of state-of-the-art ML methods combining autoencoders with contrastive learning [18]. Given the relative simplicity of the 3D OT distance, this suggests it is a promising method for anomaly detection.

Interestingly, anomaly detection based directly on the 3D OT distance outperforms methods that couple the 3D OT distance to simple machine learning algorithms such as anomaly kNN or one-class SVM. In the case of the weakly-supervised 3D OT+anomaly kNN, this may be partially attributed to the fact that the anomaly augmentations used to promote the kNN algorithm to weakly-supervised anomaly detection do not fully capture the structure of the signal models themselves. This is apparent in comparing anomaly detection with 3D OT + anomaly kNN to supervised event classification using 3D OT coupled to the kNN algorithm (see Appendix B). Here the substantially better performance of the classification task is an indication of the degree to which performance is lost by training the kNN on anomaly-augmented backgrounds rather than genuine signal.

Fully unsupervised one-class SVM is less competitive than either the 3D OT distance or 3D OT+anomaly kNN methods. As with most unsupervised methods, one-class SVMs are better at outlier detection. Namely, they focus on finding signal events in regions of low SM event probability. This is why the performance gap is smallest for the $h^\pm$ signal, which is an outlier-type signal because of the previously-noted strong total $p_T$ dependence. The other signals have a larger performance gap, indicating that they are over-density-type signals. Therefore, a simple decision boundary based on the prevalence of SM background events is less effective.

The overall performance trend of anomaly detection using the 3D OT distance — superior to OT distances coupled to simple weakly-supervised or unsupervised ML algorithms, approaching the performance of much more sophisticated ML algorithms — is largely consistent across the different signal models, with the partial exception of the $A$ signal model. Here the 3D OT distance and 3D OT+anomaly kNN are comparable to each other and weaker than autoencoder-based methods [18]. Unlike the other signal models, for the $A$ signal model the total multiplicity is a much stronger baseline discriminant than the total $p_T$. Two of the three anomaly augmentations used to construct the anomaly-augmented background dataset (which was subsequently used as a proxy for signal for training the 3D OT+anomaly kNN) capture this property. Therefore, the anomaly-augmented backgrounds are a better proxy for the true $A$ signal events. This is apparent in the smaller gap between anomaly detection using anomaly kNN and classification using kNN, and largely explains why the 3D OT+anomaly kNN is more competitive with the direct use of 3D OT distances for the $A$ signal model. The fact that both methods lag behind more sophisticated autoencoder-based methods suggests that there is additional information in signal and background events that is not captured by the simple OT-based methods presented here. We comment on directions for future directions to improve upon these results in Section 5.

In principle, OT methods may be used for anomaly detection both at trigger level and in offline analyses. There are two main considerations governing the feasibility of using OT methods in an online setting for L1 trigger-level data: memory limitations and computational evaluation time [1]. State-of-the-art ML methods are typically fast to evaluate but may require significant pruning to meet memory constraints [17]. In general, calculating the OT distance between two events is computationally costly, but many approximation strategies can lower this computational burden [36]. Moreover, the computational cost is less intense when the number of probability masses is low (which is the case for L1 trigger data), and scales very well with an increase in the dimension of the underlying ground space. The memory limitations for OT methods may also be less intense than state-of-the-art ML methods. For OT methods, the main memory requirement would be storing a representative sample of SM background events to compare a test event to. Both of these considerations (memory and computational cost) can be lessened by calculating the OT distances between a test event and the $k$-medoids of the SM background data, as was done in previous work [38, 40, 44]. Alternatively, the costs may be significantly reduced by using linearized OT [36], which provides an isometric linear embedding of all the background events into Euclidean space. Using this linearized strategy, determining the anomaly score for an event using the minimum 3D OT distance to the background sample would then be reduced to computing a single 3D OT distance to a suitably-chosen reference event, combined with the efficient calculation of Euclidean distances to events in the background sample. A quantitative study investigating the above considerations would be beneficial and is left to future work.

## 5   Conclusion

The plethora of possible extensions of the Standard Model (and the correspondingly vast space of potential signals) motivates the development of model-agnostic search techniques such as anomaly detection. In this paper, we have initiated the use of balanced OT distances for event-level anomaly detection, both on their own and coupled to interpretable, unsupervised (or weakly supervised) machine learning algorithms. In both cases, the 2-Wasserstein distance based on a three-dimensional ground space that includes $p_T/\text{GeV}$ as a coordinate alongside the physical $\eta, \phi$ calorimeter coordinates ("3D OT") outperforms the 2-Wasserstein distance using the $\eta, \phi$ coordinates alone ("2D OT"). Interestingly, simply using the minimum 3D 2-Wasserstein distance between a test event and a background sample as an anomaly score outperforms anomaly detection that couples the 2-Wasserstein distances between events to simple machine learning algorithms such as kNN ("3D OT+anomaly kNN") or one-class SVM ("3D OT+One-class SVM"). Although this simple OT distance-based approach to anomaly detection does not fully match the performance of more sophisticated ML methods on the same dataset [18], it demonstrates the key features and potential advantages of using optimal transport for anomaly detection in collider data.

There are two main directions for future work to improve upon these baselines. The first is to incorporate more information into the optimization problem. For example, one could consider adding additional ground space dimensions, e.g. multiplicity, to capitalize on known differences between SM and many BSM signals. Along these lines, one could consider encoding species-type information, i.e. multi-species OT. Ref. [40] investigated adding species difference as a weight on the ground space distance. There are a number of well-posed forms of multi-species optimal transport that may prove fruitful in this setting [35].

The second strategy for future work is to improve the correspondence between physical differences and OT distances between events. Concretely, one could achieve this by accounting for symmetry transformations of the underlying ground space. Ref. [34] investigated choosing

a ground space which is manifestly invariant to event-level physical transformations. Another straightforward, and mathematically rigorous, way to handle this would be to add ground-space transformations (e.g. rotations) to the problem as additional optimization steps [56]. It might also be interesting to consider other varieties of OT distances which are manifestly insensitive to such ground-space transformations. Namely, the Gromov-Wasserstein distance [57], originally designed to compute OT distances between distributions with ground spaces of different dimensions, only depends on the relative locations of the probability masses, $p_i$'s, within each event. It is therefore, manifestly invariant to translations and rotations of the underlying ground space. While the Gromov-Wasserstein distance itself is computationally non-trivial, it admits several approximations [58] which are computationally efficient for discrete distributions with a small number of $p_i$'s.

# Acknowledgements

We would like to thank Katy Craig and Tianji Cai for useful conversations about optimal transport, and Tianji Cai for sharing her kNN classification code. We would also like to thank Gregor Kasieczka, Cristiano Fanelli, Thomas G. Rizzo, and Michael Peskin for useful discussions. This work was performed in part at the Aspen Center for Physics, which is supported by National Science Foundation grant PHY-2210452. JNH was supported by the National Science Foundation under Grant No. NSF PHY-1748958 and by the Gordon and Betty Moore Foundation through Grant No. GBMF7392. NC was supported by the Department of Energy under Grant No. DE-SC0011702.

# A   Appendix: Baseline performance

In this appendix, we summarize the performance of common baseline methods. When evaluating these baseline methods (Total $p_T$, MET, and Total Multiplicity) we construct a test set from $1,000$ SM background events and $1,000$ BSM signal events (for each signal model). For comparison, we also include the results from a recent work studying the performance of various autoencoder methods on this dataset [18].

# B   Appendix: Comparison with Classification Performance

To investigate the information contained in the 2D vs 3D ground spaces, we looked at the performance of a supervised kNN algorithm trained to distinguish SM from each BSM model individually. This serves as a best-case performance limit for the kNN anomaly detection algorithm. Indeed we find that the 3D ground space contains much more information than its 2D counterpart. This is likely due to the importance of total transverse momentum, $p_T$, in this dataset (see Table 2). To test this, we also consider kNN algorithms trained on events with "planed" total $p_T$ [59]. We find that, in the $A$ case, the 2D ground space achieves a performance boost both compared to the un-planed case as well as the total $p_T$. This indicates that, for some signal models, there can be extra discriminating information contained in the $\eta, \phi$ distribution of an event. We also note, that for all signal cases, the 3D OT+kNN classification performance outperforms the 3D OT+kNN anomaly detection performance. This indicates, that there is still more discriminating information that could be extracted.

| | | Common Baselines | | | Recent Results from Ref. [18] | |
| --- | --- | --- | --- | --- | --- | --- |
| | | Total $p_\mathrm{T}$ | MET | Total Multiplicity | AE-Raw | AnomCLR+ |
| AUC | $A$ | 0.7409 ± 0.01102 | 0.3690 ± 0.01391 | **0.8914 ± 0.006006** | 0.885 ± 0.002 | **0.909 ± 0.003** |
| | $h^0$ | 0.7076 ± 0.008915 | 0.5977 ± 0.01464 | **0.7377 ± 0.009054** | 0.755 ± 0.002 | **0.776 ± 0.002** |
| | $h^\pm$ | **0.9257 ± 0.003988** | 0.8371 ± 0.009309 | 0.8783 ± 0.005394 | 0.900 ± 0.004 | **0.930 ± 0.001** |
| | $LQ$ | 0.8421 ± 0.006929 | 0.5609 ± 0.01077 | **0.8750 ± 0.006784** | 0.856 ± 0.002 | **0.880 ± 0.001** |
| $\epsilon_b^{-1}$ ($\epsilon_s = 0.3$) | $A$ | 13.27 ± 1.468 | 2.014 ± 0.1082 | **82.15 ± 17.13** | 47 ± 2 | **139 ± 20** |
| | $h^0$ | **19.97 ± 3.738** | 8.262 ± 1.082 | 13.56 ± 1.366 | 14.9 ± 0.7 | **23 ± 1** |
| | $h^\pm$ | 88.29 ± 17.08 | **388.9 ± 335.2** | 46.85 ± 5.803 | 60 ± 10 | **171 ± 7** |
| | $LQ$ | 19.85 ± 1.315 | 4.78 ± 0.3807 | **38.15 ± 5.172** | 24.4 ± 6 | **39 ± 1** |
| SI ($\epsilon_s = 0.3$) | $A$ | 1.091 ± 0.05907 | 0.4255 ± 0.01143 | **2.694 ± 0.2675** | 2.05 ± 0.05 | **3.5 ± 0.2** |
| | $h^0$ | **1.334 ± 0.1200** | 0.8601 ± 0.05535 | 1.103 ± 0.05745 | 1.16 ± 0.03 | **1.44 ± 0.03** |
| | $h^\pm$ | **2.793 ± 0.2746** | 5.284 ± 2.213 | 2.045 ± 0.1246 | 2.3 ± 0.2 | **3.9 ± 0.1** |
| | $LQ$ | 1.335 ± 0.04426 | 0.6552 ± 0.02631 | **1.845 ± 0.1200** | 1.48 ± 0.02 | **1.88 ± 0.02** |
| Max F1 ($\epsilon_s$) | $A$ | 0.7144 ± 0.003200 (0.86) | 0.6667 ± 0 (1) | **0.8249 ± 0.007316 (0.83)** | - | - |
| | $h^0$ | 0.6715 ± 0.008374 (0.77) | 0.6667 ± 0 (1) | **0.7212 ± 0.002918 (0.91)** | - | - |
| | $h^\pm$ | **0.8639 ± 0.004261 (0.89)** | 0.7542 ± 0.00732 (0.76) | 0.8064 ± 0.006508 (0.91) | - | - |
| | $LQ$ | **0.8639 ± 0.004261 (0.89)** | 0.6667 ± 0 (1) | 0.8117 ± 0.008232 (0.93) | - | - |
| Max SI ($\epsilon_s$) | $A$ | 1.200 ± 0.04191 (0.68) | 1.000 ± 0 (1) | **2.762 ± 0.1713 (0.42)** | - | - |
| | $h^0$ | **1.396 ± 0.2258 (0.20)** | 1.049 ± 0.1412 (0.20) | 1.167 ± 0.02742 (0.72) | - | - |
| | $h^\pm$ | 3.130 ± 0.6426 (0.20) | **8.164 ± 7.003 (0.22)** | 2.092 ± 0.09866 (0.54) | - | - |
| | $LQ$ | 1.611 ± 0.06085 (0.65) | 1.000 ± 0 (1) | **1.946 ± 0.07213 (0.51)** | - | - |

Table 2: The performance of baseline methods for each signal dataset are reported. The performance metrics considered are the area under the ROC curve (AUC), inverse background rejection rate ($\epsilon_b^{-1}$), and Significance Improvement (SI). Where the latter two are reported at a signal efficiency rate of $\epsilon_s = 0.3$. We also report the maximum SI and F1 values and their corresponding $\epsilon_s$ in the range $\epsilon_s \in [0.2, 1]$. A lower signal efficiency threshold ($\epsilon_s = 0.2$) is imposed when finding the maximum SI value, since the SI curve becomes unstable for low signal efficiencies. We only consider baseline methods which do not make reference to the identities of particles (e.g. lepton multiplicity). Even though these methods are standard [53], comparison to such quantities would be misleading since the OT methods employed do not currently have access to particle identity information. For comparison, we also include the results from a recent work studying the performance of various autoencoder methods on this dataset [18].

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

| | | 2D | OT+kNN classification performance 2D planed total $p_T$ (units are GeV) | | 3D |
|---|---|---|---|---|---|
| | | | $p_T \in (50, 100)$ | $p_T \in (500, 1000)$ | |
| AUC | $A$ | $0.6948 \pm 0.01795$ | $0.7974 \pm 0.008994$ | $0.7989 \pm 0.02231$ | $\mathbf{0.9021 \pm 0.01426}$ |
| | $h^0$ | $0.6698 \pm 0.01178$ | $0.6761 \pm 0.03569$ | $0.5829 \pm 0.02124$ | $\mathbf{0.7713 \pm 0.01814}$ |
| | $h^\pm$ | $0.8103 \pm 0.01673$ | $0.6071 \pm 0.02694$ | $0.6203 \pm 0.03418$ | $\mathbf{0.9198 \pm 0.006547}$ |
| | $LQ$ | $0.7906 \pm 0.02531$ | $0.7469 \pm 0.02149$ | $0.5285 \pm 0.01536$ | $\mathbf{0.8766 \pm 0.01415}$ |
| $\epsilon_b^{-1}$ ($\epsilon_s = 0.3$) | $A$ | $7.923 \pm 1.295$ | $32.51 \pm 6.754$ | $27.24 \pm 14.05$ | $\mathbf{111.8 \pm 44.36}$ |
| | $h^0$ | $8.172 \pm 1.359$ | $7.728 \pm 2.137$ | $4.955 \pm 0.6116$ | $\mathbf{18.25 \pm 1.911}$ |
| | $h^\pm$ | $17.49 \pm 4.276$ | $5.751 \pm 1.537$ | $6.297 \pm 1.55$ | $\mathbf{102.2 \pm 59.91}$ |
| | $LQ$ | $16.01 \pm 4.558$ | $11.44 \pm 2.975$ | $3.856 \pm 0.5763$ | $\mathbf{28.49 \pm 6.938}$ |
| SI ($\epsilon_s = 0.3$) | $A$ | $0.8413 \pm 0.06876$ | $1.699 \pm 0.1716$ | $1.513 \pm 0.3905$ | $\mathbf{3.094 \pm 0.5989}$ |
| | $h^0$ | $0.8542 \pm 0.07158$ | $0.8266 \pm 0.1075$ | $0.6663 \pm 0.04145$ | $\mathbf{1.279 \pm 0.06832}$ |
| | $h^\pm$ | $1.245 \pm 0.1467$ | $0.7138 \pm 0.08855$ | $0.7468 \pm 0.09285$ | $\mathbf{2.884 \pm 0.8718}$ |
| | $LQ$ | $1.187 \pm 0.1717$ | $1.005 \pm 0.1321$ | $0.5873 \pm 0.04454$ | $\mathbf{1.585 \pm 0.2122}$ |
| Max F1 ($\epsilon_s$) | $A$ | $0.6980 \pm 0.03021 \ (0.94)$ | $0.7487 \pm 0.01468 \ (0.86)$ | $0.7425 \pm 0.01830 \ (0.77)$ | $\mathbf{0.8352 \pm 0.01350 \ (0.84)}$ |
| | $h^0$ | $0.6845 \pm 0.01654 \ (0.92)$ | $0.6883 \pm 0.01781 \ (0.89)$ | $0.6689 \pm 0.01104 \ (0.99)$ | $\mathbf{0.7323 \pm 0.01493 \ (0.94)}$ |
| | $h^\pm$ | $0.7679 \pm 0.02457 \ (0.87)$ | $0.6676 \pm 0.01645 \ (0.98)$ | $0.6728 \pm 0.01939 \ (0.92)$ | $\mathbf{0.8569 \pm 0.01173 \ (0.88)}$ |
| | $LQ$ | $0.7501 \pm 0.02720 \ (0.89)$ | $0.7226 \pm 0.01706 \ (0.83)$ | $0.6672 \pm 0.01574 \ (0.99)$ | $\mathbf{0.8143 \pm 0.01585 \ (0.88)}$ |
| Max SI ($\epsilon_s$) | $A$ | $1.102 \pm 0.04632 \ (0.84)$ | $1.882 \pm 0.4661 \ (0.25)$ | $1.711 \pm 0.4545 \ (0.40)$ | $\mathbf{3.407 \pm 0.997 \ (0.25)}$ |
| | $h^0$ | $1.051 \pm 0.0241 \ (0.81)$ | $1.086 \pm 0.08949 \ (0.69)$ | $1.005 \pm 0.01142 \ (0.99)$ | $\mathbf{1.343 \pm 0.1165 \ (0.57)}$ |
| | $h^\pm$ | $1.497 \pm 0.1156 \ (0.75)$ | $1.003 \pm 0.005958 \ (0.98)$ | $1.021 \pm 0.03324 \ (0.90)$ | $\mathbf{2.96 \pm 0.6393 \ (0.46)}$ |
| | $LQ$ | $1.402 \pm 0.146 \ (0.60)$ | $1.256 \pm 0.04621 \ (0.67)$ | $1.002 \pm 0.005666 \ (0.99)$ | $\mathbf{1.957 \pm 0.2496 \ (0.64)}$ |

Table 3: Classification performance of kNN+OT method for background against each signal model. The performance metrics considered are the area under the ROC curve (AUC), inverse background efficiency rate ($\epsilon_b^{-1}$), and Significance Improvement (SI). Where the latter two are reported at a signal efficiency rate of $\epsilon_s = 0.3$. We also report the maximum SI and F1 values and their corresponding $\epsilon_s$ in the range $\epsilon_s \in [0.25, 1]$. A lower signal efficiency threshold ($\epsilon_s = 0.25$) is imposed when finding the maximum SI value, since the SI curve becomes unstable for low signal efficiencies. Note that for the planed total $p_T$ cases we chose a representative low and high range.

[6] G. Aad *et al.*, *Anomaly detection search for new resonances decaying into a Higgs boson and a generic new particle X in hadronic final states using $\sqrt{s} = 13$ TeV pp collisions with the ATLAS detector* (2023), 2306.03637.

[7] G. Aad *et al.*, *Dijet resonance search with weak supervision using $\sqrt{s} = 13$ TeV pp collisions in the ATLAS detector*, Phys. Rev. Lett. **125**(13), 131801 (2020), doi:10.1103/PhysRevLett.125.131801, 2005.02983.

[8] E. Govorkova *et al.*, *Autoencoders on field-programmable gate arrays for real-time, unsupervised new physics detection at 40 MHz at the Large Hadron Collider*, Nature Mach. Intell. **4**, 154 (2022), doi:10.1038/s42256-022-00441-3, 2108.03986.

[9] N. Zipper, *Testing a Neural Network for Anomaly Detection in the CMS Global Trigger Test Crate during Run 3*, In *Topical Workshop on Electronics for Particle Physics* (2023), 2312.10009.

[10] M. W. Asres, C. W. Omlin, L. Wang, D. Yu, P. Parygin, J. Dittmann, G. Karapostoli, M. Seidel, R. Venditti, L. Lambrecht, E. Usai, M. Ahmad *et al.*, *Spatio-temporal anomaly detection with graph networks for data quality monitoring of the hadron calorimeter* (2023), 2311.04190.

[11] G. Grosso, N. Lai, M. Migliorini, J. Pazzini, A. Triossi, M. Zanetti and A. Zucchetta, *Triggerless data acquisition pipeline for Machine Learning based statistical anomaly detection* (2023), 2311.02038.

[12] D. Abadjiev *et al.*, *Autoencoder-based Anomaly Detection System for Online Data Quality Monitoring of the CMS Electromagnetic Calorimeter* (2023), 2309.10157.

[13] A. Harilal, K. Park, M. Andrews and M. Paulini, *Autoencoder-based Online Data Quality Monitoring for the CMS Electromagnetic Calorimeter*, In *21th International Workshop on Advanced Computing and Analysis Techniques in Physics Research: AI meets Reality* (2023), 2308.16659.

[14] S. V. Chekanov and R. Zhang, *Boosting sensitivity to new physics with unsupervised anomaly detection in dijet resonance search* (2023), 2308.02671.

[15] B. Nachman, *Anomaly Detection for Physics Analysis and Less than Supervised Learning* (2020), 2010.14554.

[16] HEP ML Community, *A Living Review of Machine Learning for Particle Physics*.

[17] V. Belis, P. Odagiu and T. K. Årrestad, *Machine Learning for Anomaly Detection in Particle Physics* (2023), 2312.14190.

[18] B. M. Dillon, L. Favaro, F. Feiden, T. Modak and T. Plehn, *Anomalies, Representations, and Self-Supervision* (2023), 2301.04660.

[19] J. Schuhmacher, L. Boggia, V. Belis, E. Puljak, M. Grossi, M. Pierini, S. Vallecorsa, F. Tacchino, P. Barkoutsos and I. Tavernelli, *Unravelling physics beyond the standard model with classical and quantum anomaly detection* (2023), 2301.10787.

[20] T. Heimel, G. Kasieczka, T. Plehn and J. M. Thompson, *QCD or What?*, SciPost Phys. **6**(3), 030 (2019), doi:10.21468/SciPostPhys.6.3.030, 1808.08979.

[21] M. Farina, Y. Nakai and D. Shih, *Searching for New Physics with Deep Autoencoders*, Phys. Rev. D **101**(7), 075021 (2020), doi:10.1103/PhysRevD.101.075021, 1808.08992.

[22] J. Batson, C. G. Haaf, Y. Kahn and D. A. Roberts, *Topological Obstructions to Autoencoding*, JHEP **04**, 280 (2021), doi:10.1007/JHEP04(2021)280, 2102.08380.

[23] D. Gong, L. Liu, V. Le, B. Saha, M. R. Mansour, S. Venkatesh and A. van den Hengel, *Memorizing normality to detect anomaly: Memory-augmented deep autoencoder for unsupervised anomaly detection* (2019), 1904.02639.

[24] T. Finke, M. Krämer, A. Morandini, A. Mück and I. Oleksiyuk, *Autoencoders for unsupervised anomaly detection in high energy physics*, JHEP **06**, 161 (2021), doi:10.1007/JHEP06(2021)161, 2104.09051.

[25] B. M. Dillon, T. Plehn, C. Sauer and P. Sorrenson, *Better Latent Spaces for Better Autoencoders*, SciPost Phys. **11**, 061 (2021), doi:10.21468/SciPostPhys.11.3.061, 2104.08291.

[26] B. M. Dillon, L. Favaro, T. Plehn, P. Sorrenson and M. Krämer, *A Normalized Autoencoder for LHC Triggers* (2022), 2206.14225.

[27] M. Freytsis, M. Perelstein and Y. C. San, *Anomaly Detection in Presence of Irrelevant Features* (2023), 2310.13057.

[28] C. Fanelli and J. Giroux, *ELUQuant: Event-Level Uncertainty Quantification in Deep Inelastic Scattering* (2023), 2310.02913.

[29] T. Finke, M. Hein, G. Kasieczka, M. Krämer, A. Mück, P. Prangchaikul, T. Quadfasel, D. Shih and M. Sommerhalder, *Back To The Roots: Tree-Based Algorithms for Weakly Supervised Anomaly Detection* (2023), 2309.13111.

[30] T. Golling, G. Kasieczka, C. Krause, R. Mastandrea, B. Nachman, J. A. Raine, D. Sengupta, D. Shih and M. Sommerhalder, *The Interplay of Machine Learning–based Resonant Anomaly Detection Methods* (2023), 2307.11157.

[31] G. Aad *et al.*, *Search for new phenomena in two-body invariant mass distributions using unsupervised machine learning for anomaly detection at $\sqrt{s} = 13$ TeV with the ATLAS detector* (2023), 2307.01612.

[32] V. Mikuni and B. Nachman, *High-dimensional and Permutation Invariant Anomaly Detection* (2023), 2306.03933.

[33] C. Fanelli, J. Giroux and Z. Papandreou, *'Flux+Mutability': a conditional generative approach to one-class classification and anomaly detection*, Mach. Learn. Sci. Tech. **3**(4), 045012 (2022), doi:10.1088/2632-2153/ac9bcb, 2204.08609.

[34] A. J. Larkoski and J. Thaler, *A spectral metric for collider geometry*, JHEP **08**, 107 (2023), doi:10.1007/JHEP08(2023)107, 2305.03751.

[35] T. Cai, J. Cheng, K. Craig and N. Craig, *Which metric on the space of collider events?*, Phys. Rev. D **105**(7), 076003 (2022), doi:10.1103/PhysRevD.105.076003, 2111.03670.

[36] T. Cai, J. Cheng, N. Craig and K. Craig, *Linearized optimal transport for collider events*, Phys. Rev. D **102**(11), 116019 (2020), doi:10.1103/PhysRevD.102.116019, 2008.08604.

[37] P. T. Komiske, E. M. Metodiev and J. Thaler, *The Hidden Geometry of Particle Collisions*, JHEP **07**, 006 (2020), doi:10.1007/JHEP07(2020)006, 2004.04159.

[38] P. T. Komiske, E. M. Metodiev and J. Thaler, *Metric Space of Collider Events*, Phys. Rev. Lett. **123**(4), 041801 (2019), doi:10.1103/PhysRevLett.123.041801, 1902.02346.

[39] M. Algren, J. A. Raine and T. Golling, *Decorrelation using Optimal Transport* (2023), 2307.05187.

[40] M. Crispim Romão, N. F. Castro, J. G. Milhano, R. Pedro and T. Vale, *Use of a generalized energy Mover's distance in the search for rare phenomena at colliders*, Eur. Phys. J. C **81**(2), 192 (2021), doi:10.1140/epjc/s10052-021-08891-6, 2004.09360.

[41] A. Davis, T. Menzo, A. Youssef and J. Zupan, *Earth mover's distance as a measure of CP violation*, JHEP **06**, 098 (2023), doi:10.1007/JHEP06(2023)098, 2301.13211.

[42] D. Ba, A. S. Dogra, R. Gambhir, A. Tasissa and J. Thaler, *SHAPER: can you hear the shape of a jet?*, JHEP **06**, 195 (2023), doi:10.1007/JHEP06(2023)195, 2302.12266.

[43] T. Gaertner and J. Reiten, *Unsupervised learning in the metric space of jets* (2023), 2312.06948.

[44] K. Fraser, S. Homiller, R. K. Mishra, B. Ostdiek and M. D. Schwartz, *Challenges for unsupervised anomaly detection in particle physics*, JHEP **03**, 066 (2022), doi:10.1007/JHEP03(2022)066, 2110.06948.

[45] S. E. Park, P. Harris and B. Ostdiek, *Neural embedding: learning the embedding of the manifold of physics data*, JHEP **07**, 108 (2023), doi:10.1007/JHEP07(2023)108, 2208.05484.

[46] B. Schölkopf, R. C. Williamson, A. Smola, J. Shawe-Taylor and J. Platt, *Support vector method for novelty detection*, Advances in neural information processing systems **12** (1999).

[47] T. Cover and P. Hart, *Nearest neighbor pattern classification*, IEEE Transactions on Information Theory **13**(1), 21 (1967), doi:10.1109/TIT.1967.1053964.

[48] G. Kasieczka, R. Mastandrea, V. Mikuni, B. Nachman, M. Pettee and D. Shih, *Anomaly detection under coordinate transformations*, Phys. Rev. D **107**(1), 015009 (2023), doi:10.1103/PhysRevD.107.015009, 2209.06225.

[49] T. Sjöstrand, S. Ask, J. R. Christiansen, R. Corke, N. Desai, P. Ilten, S. Mrenna, S. Prestel, C. O. Rasmussen and P. Z. Skands, *An introduction to PYTHIA 8.2*, Comput. Phys. Commun. **191**, 159 (2015), doi:10.1016/j.cpc.2015.01.024, 1410.3012.

[50] J. de Favereau, C. Delaere, P. Demin, A. Giammanco, V. Lemaître, A. Mertens and M. Selvaggi, *DELPHES 3, A modular framework for fast simulation of a generic collider experiment*, JHEP **02**, 057 (2014), doi:10.1007/JHEP02(2014)057, 1307.6346.

[51] M. Cacciari, G. P. Salam and G. Soyez, *FastJet User Manual*, Eur. Phys. J. C **72**, 1896 (2012), doi:10.1140/epjc/s10052-012-1896-2, 1111.6097.

[52] M. Cacciari, G. P. Salam and G. Soyez, *The anti-$k_t$ jet clustering algorithm*, JHEP **04**, 063 (2008), doi:10.1088/1126-6708/2008/04/063, 0802.1189.

[53] V. Mikuni, B. Nachman and D. Shih, *Online-compatible unsupervised nonresonant anomaly detection*, Phys. Rev. D **105**(5), 055006 (2022), doi:10.1103/PhysRevD.105.055006, 2111.06417.

[54] R. Flamary, N. Courty, A. Gramfort, M. Z. Alaya, A. Boisbunon, S. Chambon, L. Chapel, A. Corenflos, K. Fatras, N. Fournier, L. Gautheron, N. T. Gayraud *et al.*, *Pot: Python optimal transport*, Journal of Machine Learning Research **22**(78), 1 (2021).

[55] F. Pedregosa, G. Varoquaux, A. Gramfort, V. Michel, B. Thirion, O. Grisel, M. Blondel, P. Prettenhofer, R. Weiss, V. Dubourg, J. Vanderplas, A. Passos *et al.*, *Scikit-learn: Machine learning in Python*, Journal of Machine Learning Research **12**, 2825 (2011).

[56] T. Cai, K. Craig, N. Craig and X. Lin, *to appear* (2024).

[57] F. Mémoli, *Gromov–wasserstein distances and the metric approach to object matching*, Foundations of computational mathematics **11**, 417 (2011).

[58] M. Li, J. Yu, H. Xu and C. Meng, *Efficient approximation of gromov-wasserstein distance using importance sparsification* (2023), 2205.13573.

[59] S. Chang, T. Cohen and B. Ostdiek, *What is the Machine Learning?*, Phys. Rev. D **97**(5), 056009 (2018), doi:10.1103/PhysRevD.97.056009, 1709.10106.