# Peer review of "Exploring Optimal Transport for Event-Level Anomaly Detection at the Large Hadron Collider"

_SciPost Physics_

## Round 2 · Referee Report · Anonymous (Referee 1) · 2024-8-9

Report

This manuscript considers the timely topic of anomaly detection at the Large Hadron Collider (LHC), and proposes using optimal transport (OT) techniques to define anomaly scores. Of the four studied methods, the overall best performance was obtained by using a "3D" ground metric and assessing the minimum OT distance between a test event and an ensemble of N standard model (SM) reference events. This study is based on a well-established benchmark LHC dataset, and the authors compare the performance of their method to simple observable-based cuts and to other machine-learning based strategies.

With a few revisions/clarifications, this manuscript would be suitable for SciPost Physics Core, since it is a well-written document that introduces a new technique for anomaly detection. To qualify for SciPost Physics, though, a number of conceptual and computational issues would need to be expounded on further.

Let me start with the minimum updates that would be needed for this manuscript to meet the standards for SciPost Physics Core. These updates would help clarify the existing results in the paper. Roughly in the order these issues appear in the text:

  1. For the four signal processes introduced in Section 2, it would be helpful to know their cross sections relative to the standard model backgrounds. This would give the reader a sense of the level of background rejection (or integrated luminosity) needed to see these signals.

  2. I couldn't find anywhere in the text where it says what number N of SM reference events are actually used in the minimum-OT-to-an-ensemble criteria. The text talks about using 1000 SM test events, but if I understand correctly, test events are different from reference events. I assume that 1000 reference events were also used, but that would be good to clarify.

  3. The authors talk about OT on the eta/phi "plane", but because of periodic boundary conditions, one should really talk about the eta/phi "cylinder". The authors should clarify whether they use "arc lengths" or "chord lengths" (or something else) on this cylinder to define the ground metric. I assume that arc lengths are used (i.e. treat the cylinder like a flat plane with periodic boundary conditions in the phi direction).

  4. For the "3D" case, the authors talk about non-existent objects living at the "origin" since they are assigned zero pT. Given the question above about the eta/phi space being a cylinder, where exactly is the "origin"? I assume that the authors are referring to a cylindrical geometry where the origin has pT=0 and eta=0 but phi is not defined. If that's the case, then the natural distance in that geometry are "chord-based" (i.e. straight lines in the cylindrical embedding space), which the authors should also clarify.

  5. There are various places where the authors do a scan to determine hyperparameter values (like the k of k-nearest neighbors and the gamma for one-class support vector machines). Could the authors say what values of the hyperparameters were ultimately used?

  6. The supervised kNN results are mentioned, but as far as I can tell, Table 3 with those results is not actually referenced as "Table 3" anywhere in the text. Can that be fixed?

  7. The OT results will depend on the choice of SM reference events (assuming N isn't defined as the entire SM background sample). The authors define an uncertainty band from using 5 separate test sets, but they do not seem to assess the uncertainty from using different sets of SM reference events. Either the spread from using different reference events should be included, or a justification should be given for why those uncertainties are expected to be small/irrelevant.

  8. In Figure 1, the authors say that the ROC curve for the one-class SVM case is "trivial". I'm not sure what the authors mean by this, since if the ROC curve is trivial, then how do they define an AUC (area under the ROC curve)? My guess is that the one-class SVM case corresponds to a point on the ROC plane, and the actual ROC curve arises from connecting that point to the (0,1) and (1,0) end points. It would be helpful to show that curve (or at least the point) on the plot, to be able to visually compare it to the other three methods.

For consideration in SciPost Physics, the authors would need to address a number of conceptual and computational issues. SciPost Physics is aimed at publishing high-impact results, and while the proof-of-concept OT results in this manuscript are interesting, they leave unanswered a number of questions that would need to be addressed to meet this journal's standards.

A. Putting Table 1 and Table 2 side by side, it seems that the OT methods (Table 1) do not provide much improvement (if at all) over the common observable-based baselines (Table 2). This is not necessarily a problem, since there no universal way to compare different anomaly detection strategies. Still, given that the authors anticipate potentially running their algorithm in the L1 trigger, what justification can the authors give to LHC experimentalists to implement a computational expensive OT approach when simple cuts yield comparable performance on these benchmark problems?

B. Another way of asking the question about is whether, given the same information, should one apply an observable-based or OT-based analysis strategy? For each of the baseline observables considered in Table 2, it would be straightforward to consider an OT variant, where one constructs a 1D ground metric built from total pT, MET, or total multiplicity (or maybe even a 3D ground metric combining the three), and then applies the same minimum-OT-to-an-ensemble philosophy. Of course, this is a bit of overkill, since each event only has one total pT value, so the full machinery of OT isn't really needed to compare pT values between events. Still, if one sees better performance from an OT-style approach even with standard observables, that would help underline the value of the minimum-OT-to-an-ensemble approach. (And if one doesn't see better performance, it would give the authors a chance to explain why not.)

C. The authors say that the 3D metric outperforms the 2D metric because it doesn't include pT information. While I certainly understand the reasons why the authors don't want to consider unbalanced optimal transport, it is straightforward to include pT information even with the 2D metric and still do balanced OT. For example, if the authors are using chord distances on the cylinder for the 2D case (i.e. embedding the 2D cylinder into a 3D space and using 3D distances), then one easy strategy is to represent the pT imbalance as a particle at the center of the cylinder. Given that the authors already represent non-existent objects as particles at the "origin" in the 3D case, it would make sense to do a similar approach in the 2D case by putting the pT-imblance at the "center" of the cylinder.

D. The authors use the minimum OT to a set of N reference events in their analysis. N is an important hyperparameter, but the authors do not specify the value of N they use (as mentioned in point 2 above) nor do they consider how their results might change from different values of N. Either a scan over N should be performed, or a reason why the choice of N doesn't matter should be given.

E. Taking the minimum distance to N events is quite delicate, since for any individual test event, one is highly sensitive to the precise choice of reference events. There are very few regions of phase space that the SM can't populate with some non-zero probability, so there is a chance (albeit small) that one of the reference SM events is "accidentally" close to the test event. Said another way, the value of the minimum is sensitive to tails in the phase space distribution. To guard against these fluctuations in the choice of reference events, the authors should consider more stable ways to process the ensemble. They already mention the possibility of using k-medoids. Another stable option (which is more inline with their minimum philosophy) is to take the smallest 1% or 0.1% OT value (in the quantile sense). It would be interesting to study how performance changes as this quantile fraction changes.

F. Related to the above point, the authors mention that the one-class SVM is not well suited for over-density-type signals. In the N goes to infinity limit of the minimum-OT-to-an-ensemble approach, isn't that basically the same as one-class classification? My reasoning is that if there is any probability for the SM to populate a region of phase space, then the minimum OT distance would be zero. In this sense, it appears that finite N acts like a regulator, whose value could have a large impact on performance.

G. The authors hint at the high computational cost of OT, but they don't provide any information on the runtime of their method in its current form or the required runtime to make this practical for L1 trigger implementation.

I hope that the authors engage with this longer list of questions, both to satisfy the SciPost Physics criteria and to satisfy this referee's curiosity. OT for anomaly detection seems like a potentially powerful method, and additional studies would help clarify how and why it is effective.

Recommendation

Ask for major revision

---

## Round 2 · Referee Report · Anonymous (Referee 2) · 2024-9-10

Report

The authors performed a quantitative analysis of optimal transport as an anomaly detection score using the CMS anomaly detection dataset. They showed that OT can be used for anomaly detection, albeit with a performance comparable to other methods. The paper contains interesting results and in principle can be considered for publication in SciPost, after revisions. Given the rather detailed report from Referee 1, I will not repeat any issues already raised in that report, and only mention a few points that I have identified in addition:

  1. On page 2 the authors state that “Unlike density estimation strategies [15], OT does not need to learn a probability model of the data since it is predisposed to detect distributional differences.” However, this is not entirely correct, since in order to properly interpret the OT score one in effect does need to learn a probability model of the data, albeit in a highly transformed form, translating this into distribution of OT distances. The authors should rephrase that statement to make clear to the reader that OT scores cannot be simply used out of the box, since the distribution of OT values is not known from the outset (unlike for instance chi^2, which will follow a chi^2 distribution).

  2. The authors should give more details on how classification was done using using 2D and 3D OT distances. That is, they should explicitly state on p. 6 what the test set they constructed is.

  3. The authors should discuss on how the OT based anomaly detection will scale with number of events. The minimal OT distance will decrease with growing number of events, though the statistical significance of an excess of BSM events should grow with increased luminosity (for fixed cross sections). Does OT based anomaly detection follow this expectation?

List of typographical errors:

  • p. 7: descision -> decision

  • p. 8: the the

Recommendation

Ask for major revision

---

## Round 2 · Referee Report · Anonymous (Referee 3) · 2024-10-26

Strengths

(1) original (2) rather timely (3) powerful new method of combining W2 distances with machine learning algorithm (4) clearly demonstrating that OT can outperform machine learning codes in certain cases.(5)The work has a strong potential to develop into a mainstream data-analysis in particle physics.

Weaknesses

I shall not repeat the modifications asked by the other two referees. My main issue with the paper is that the algorithm is not clearly presented and the complexity of the algorithm wrt other existing eg machine-learning methods is not properly discussed. This can be easily remedies by a flowchart (often used by computer scientists for this precise purpose).

Report

This is a nice and original piece of work and presents yet another area where Optimal Transport theory, a fast growing field, can be applied. The paper compares and combines OT with machine learning and the computational gains presented are impressive.

Requested changes

I find the algorithm used rather unclear. They use OT python package from Flamary et al (I believe ?!) and the authors also feed W2 distances to the machine learning algorithm. The complexity of the algorithlm and the flow is not clear. The speed might not be an issue for their dataset but this can easily become an important hurddle for very large particle physics data. I suggest authors make a flow chart clealry showing how their algorithm flows and mention specfically complexity and gains in using for example a different OT algorithm or machine learig method. Such flow charts are often used when presenting new or combined algorithms. It would help the reader and also demonstrate clearly the potentials of this innovative approach.

Recommendation

Publish (easily meets expectations and criteria for this Journal; among top 50%)

---

## Editorial Decision

awaiting_resubmission